# The kinase LYK5 is a major chitin receptor in *Arabidopsis* and forms a chitin-induced complex with related kinase CERK1

Yangrong Cao[1,2][†], Yan Liang[1,2][†], Kiwamu Tanaka[1,2][‡], Cuong T Nguyen[1,2], Robert P Jedrzejczak[3], Andrzej Joachimiak[3], Gary Stacey[1,2]*

[1]Division of Plant Sciences, National Center for Soybean Biotechnology, University of Missouri, Columbia, United States; [2]Department of Biochemistry, University of Missouri, Columbia, United States; [3]Midwest Center for Structural Genomics, Argonne National Laboratory, Argonne, United States

**Abstract** Chitin is a fungal microbe-associated molecular pattern recognized in *Arabidopsis* by a lysin motif receptor kinase (LYK), AtCERK1. Previous research suggested that AtCERK1 is the major chitin receptor and mediates chitin-induced signaling through homodimerization and phosphorylation. However, the reported chitin binding affinity of AtCERK1 is quite low, suggesting another receptor with high chitin binding affinity might be present. Here, we propose that AtLYK5 is the primary chitin receptor in *Arabidopsis*. Mutations in AtLYK5 resulted in a significant reduction in chitin response. However, AtLYK5 shares overlapping function with AtLYK4 and, therefore, *Atlyk4/Atlyk5-2* double mutants show a complete loss of chitin response. AtLYK5 interacts with AtCERK1 in a chitin-dependent manner. Chitin binding to AtLYK5 is indispensable for chitin-induced AtCERK1 phosphorylation. AtLYK5 binds chitin at a much higher affinity than AtCERK1. The data suggest that AtLYK5 is the primary receptor for chitin, forming a chitin inducible complex with AtCERK1 to induce plant immunity.

*For correspondence: Staceyg@ missouri.edu

[†]These authors contributed equally to this work

**Present address:** [‡]Department of Plant Pathology, Washington State University, Pullman, United States

**Competing interests:** The authors declare that no competing interests exist.

**Reviewing editor**: Thorsten Nürnberger, Tübingen University, Germany

## Introduction

As sessile organisms, plants have developed effective immune systems to defend against invading pathogens. Pathogen perception in plants can be divided into two different layers (*Jones and Dangl, 2006*). The initial response, mediated by perception of microbe-associated molecular patterns (MAMPs), is termed MAMP-triggered immunity (MTI) (*Boller and Felix, 2009*; *Macho and Zipfel, 2014*). MTI is characterized by a wide range of physiological responses, including production of reactive oxygen species (ROS), mitogen activated protein kinase (MAPK) phosphorylation, calcium influx, ion channel activation, callose deposition, growth inhibition, and expression of defense-related genes (*Macho and Zipfel, 2014*). However, adapted pathogens can inhibit MTI through the secretion of specific effector proteins or small RNAs into the cell (*Weiberg et al., 2013*, *2014*). In response, plants have evolved polymorphic nucleotide-binding site leucine-rich repeat (NBS-LRR) proteins that either directly or indirectly recognize these effectors; thereby, restoring plant immunity. This form of immunity is termed effector-triggered immunity (ETI) (*Bent and Mackey, 2007*; *Boller and Felix, 2009*). Compared to MTI, ETI is much stronger and often associated with a hypersensitive response (HR), which involves programmed cell death (*Jones and Dangl, 2006*).

Plants use cell-surface localized pattern-recognition receptors (PRRs) to detect MAMPs activating MTI (*Boller and Felix, 2009*; *Macho and Zipfel, 2014*). In plants, several PRRs have been well characterized, including FLAGELLIN-SENSING 2 (FLS2) and elongation factor-TU (EF-Tu) RECEPTOR (EFR), which detect bacterial flagellin and EF-Tu, respectively (*Gomez-Gomez and Boller, 2000*, *2002*;

**eLife digest** Invading fungi are responsible for many of the plant diseases that affect global crop production. Plants have to be able to identify these fungi, and activate the right defense strategies if they are to protect themselves. Chitin is a polymer that is found in the cell walls of all fungi, but not in plants, so if the plant detects chitin, it knows that a potentially harmful fungus may be nearby.

The detection of chitin, and the resulting activation of a plant's defenses, requires a receptor protein called CERK1. In rice, CERK1 needs to interact with another receptor protein called CEBiP, which binds to chitin. However, in *Arabidopsis thaliana*—which is widely studied in plant research—CERK1 can bind to chitin on its own, although this interaction is very weak, so it has been suggested that a second protein may be involved.

Cao et al. have now found that a receptor protein called LYK5, which is very similar to CERK1, is much better at attaching to chitin in *A. thaliana*. It can also bind to CERK1, but only when chitin is present, and is required for activation of basic plant defenses. The experiments suggest that LYK5 detects chitin on behalf of CERK1, in a similar way to how CEBiP works in rice.

The next step in this research is to work out how CERK1 and LYK5 are able to activate plant defenses.

*Zipfel et al., 2006*). These two receptors belong to the leucine rich repeat receptor like protein kinase (LRR-RLK) family (*Shiu and Bleecker, 2001*). Activation of these receptors by ligand binding induces the association with BRASSINOSTEROID INSENSITIVE 1 (BRI1)- Associated receptor Kinase 1 (BAK1) and phosphorylation of Botrytis-Induced Kinase (BIK1) (*Chinchilla et al., 2007*; *Heese et al., 2007*; *Lu et al., 2010*; *Roux et al., 2011*; *Zhang et al., 2010*). Phosphorylated BIK1 dissociates from the receptor and subsequently phosphorylates the respiratory burst oxidase homolog D (RBOHD) protein, which controls ROS production in a calcium-independent manner (*Kadota et al., 2014*; *Li et al., 2014*). Other MAMPs, such as the oligosaccharides bacterial peptidoglycan (PGN) and fungal chitin [degree of polymerization (dp) ≥ 6], are detected by lysin-motif (LysM) containing proteins. The chitin receptor was first reported in rice with the identification of the chitin-elicitor binding protein (CEBiP) (*Kaku et al., 2006*), which contains an extracellular LysM motif and a transmembrane domain, but lacks an intracellular kinase domain. Data indicate that CEBiP forms a complex with the rice chitin-elicitor receptor kinase 1 (OsCERK1) to mediate MTI in response to chitin (*Hayafune et al., 2014*; *Shimizu et al., 2010*). OsCERK1 has an active, intracellular kinase domain. The data suggest that OsCERK1 does not bind chitin but its intracellular kinase domain is activated by chitin binding to OsCEBiP (*Hayafune et al., 2014*). In *Arabidopsis thaliana*, AtCERK1 was shown to be a key chitin receptor involved in chitin perception (*Miya et al., 2007*; *Wan et al., 2008*). For example, *Atcerk1* mutant plants completely lose the ability to respond to chitin elicitation. There are three homologs of CEBiP in *Arabidopsis*, but a triple knock-out mutant, *Atlym1/Atlym2/Atlym3* lacking these three proteins was fully competent to respond to chitin treatment (*Wan et al., 2012*). However, two CEBiP like proteins appear to function in conjunction with CERK1 to mediate recognition of bacterial PGN (OsLYP4 and OsLYP6 in rice, AtLYM1 and AtLYM3 in *Arabidopsis*) (*Liu et al., 2012a*; *Willmann et al., 2011*). Another CEBiP-like protein, AtLYM2, was demonstrated to act independently of AtCERK1 to mediate chitin-induced suppression of intracellular flux through plasmodesmata (*Faulkner et al., 2013*).

In *Arabidopsis*, there are five members of the lysin-motif receptor like kinase family (LYKs), that is, AtCERK1/LysM RLK1/AtLYK1, and AtLYK2-5 (*Wan et al., 2012*). AtCERK1 was reported as the primary chitin receptor based on the mutant phenotype (*Miya et al., 2007*; *Wan et al., 2008*) but also the fact that the protein can be precipitated by binding to chitin beads (*Iizasa et al., 2010*; *Petutschnig et al., 2010*). The X-ray crystal structure of the ectodomain of AtCERK1 was solved by *Liu et al. (2012b)*. The structure predicted interaction of chitin oligomers with the second LysM motif in the extracellular domain. These authors suggested a model by which long chain chitin oligomers (dp ≥ 6) bound to the LysM domains on two monomers, resulting in homodimerization of AtCERK1. This dimerization was shown to activate the intracellular kinase domain (*Liu et al., 2012b*; *Petutschnig et al., 2010*). However, there remains the possibility that, similar to the situation in rice, the active chitin receptor is composed of more than one protein. For example, mutations in *AtLYK4* were shown to significantly

reduce the plant response to chitin (**Wan et al., 2012**), although the phenotype was not as pronounced as that of *Atcerk1* mutant plants. While the X-ray crystal structure of the ectodomain of AtCERK1 provided evidence that it is indeed a chitin binding protein, a puzzling aspect of this work is the low binding affinity (chitooctaose, $K_d$ = 45 μM) reported, based on calorimetry (**Liu et al., 2012b**). Another puzzling aspect is that mutations in AtCERK1, predicted to block chitin binding (AtCERK1[A138H]), did not block chitin-induced AtCERK1[A138H] phosphorylation (**Liu et al., 2012b**). These data led us to consider the possibility that a second protein may be involved that mediates high affinity chitin binding and works with AtCERK1 to activate MTI.

In this study, we show that mutations in AtLYK5 result in a significant reduction in the plant chitin response. AtLYK5 is required for chitin-induced AtCERK1 homodimerization and phosphorylation. AtLYK5 binds to chitin with a much higher affinity than AtCERK1. The data suggest that AtLYK5 is the primary receptor for chitin, forming a chitin-inducible complex with AtCERK1 to induce plant innate immunity.

## Results

### AtLYK5 is essential for the chitin response in *Arabidopsis*

*Arabidopsis* has five LysM receptor kinases (LYKs) (**Figure 1—figure supplement 1**). Therefore, plants mutated in each of these genes were tested for their ability to induce reactive oxygen species (ROS) in response to chitin elicitation. As expected from previous publications, mutations in *AtCERK1* showed strongly reduced ROS production (**Miya et al., 2007**; **Wan et al., 2008**), while mutations in *AtLYK4* also showed a slight reduction in ROS production upon chitin elicitation (**Wan et al., 2012**). In previous publications, which involved screening *Atlyk1-5* mutants, we reported that a transposon insertion in *AtLYK5* did not affect chitin-induced MTI (**Wan et al, 2008**, **2012**). This conclusion was based on measuring *AtWRKY53* expression upon chitin addition. At the time of these studies, the only *Atlyk5* mutant available was in the Lansberg (Ler) background (*Atlyk5-1*). As part of a new round of screening, we again examined the chitin response of *Atlyk5-1* mutant plants. qRT-PCR analysis showed that chitin treatment induced similar expression of *AtWRKY53* in both Ler wild-type and *Atlyk5-1* mutant plants (**Figure 1—figure supplement 2**); data consistent with the previously published results (**Wan, et al, 2008**, **2012**). However, in contrast to these results, the expression of *AtWRKY33* 15 min after chitin treatment was significantly reduced in *Atlyk5-1* mutant plants relative to Ler wild-type plants (**Figure 1—figure supplement 2**). Chitin-triggered MAP kinase (MPK) phosphorylation was also significantly reduced in *Atlyk5-1* mutant plants compared with Ler wild-type plants (**Figure 1—figure supplement 2**). The phosphorylated AtCERK1 triggered by chitin elicitation can be detected as a band shift based on immunoblots using anti-AtCERK1 antibody (**Figure 1—figure supplement 2**) (**Liu et al., 2012b**; **Petutschnig et al., 2010**). Chitin-triggered AtCERK1 phosphorylation was detected in Ler wild-type plants but was reduced in *Atlyk5-1* mutant plants (**Figure 1—figure supplement 2**). In general, based on chitin-triggered ROS production, Ler wild-type plants showed a lower response to chitin than Col-0 plants, while *Atlyk5-1* mutant plants showed similar ROS production to the wild-type when treated with chitin (**Figure 1—figure supplement 2**). Taken together, these experiments suggested that our original conclusion concerning AtLYK5 may not be correct; that is, this protein may be involved in chitin response. What is clear is that the *Atlyk5-1* mutant, with a transposon insertion in the 3′ region of the gene, does not exhibit a strong phenotype under all conditions. The analysis of the chitin response in the Ler ecotype is further complicated by the generally weak response to chitin elicitation.

Given these concerns, we identified and characterized a Col-0 *Atlyk5* mutant (*Atlyk5-2*) from the SALK population (**Figure 1—figure supplement 3**). This line has a T-DNA in the extracellular domain of AtLYK5 (**Figure 1—figure supplement 3**). It should be noted that in our original publications (**Wan et al., 2008**, **2012**), all of the *Atlyk* mutants, with the exception of *Atlyk5-1*, were derived from the Col-0 ecotype. As shown in **Figure 1A**, chitin-induced ROS production was significantly lower in the *Atlyk5-2* mutant plants compared to Col-O wild-type plants (**Figure 1A**). Calcium influx is activated by exposure of wild-type plants to chitin. Similar treatment of *Atlyk5-2* mutant plants showed a 90% reduction and a significant delay in the calcium response, while *Atcerk1* mutant plants showed essentially no calcium response to chitin (**Figure 1B**). Flagellin-triggered calcium influx in both *Atcerk1* and *Atlyk5-2* mutant plants were similar to Col-0 wild-type plants (**Figure 1—figure supplement 4**), indicating that these defects were specific to chitin and not to a general effect on MTI. MPK3 and

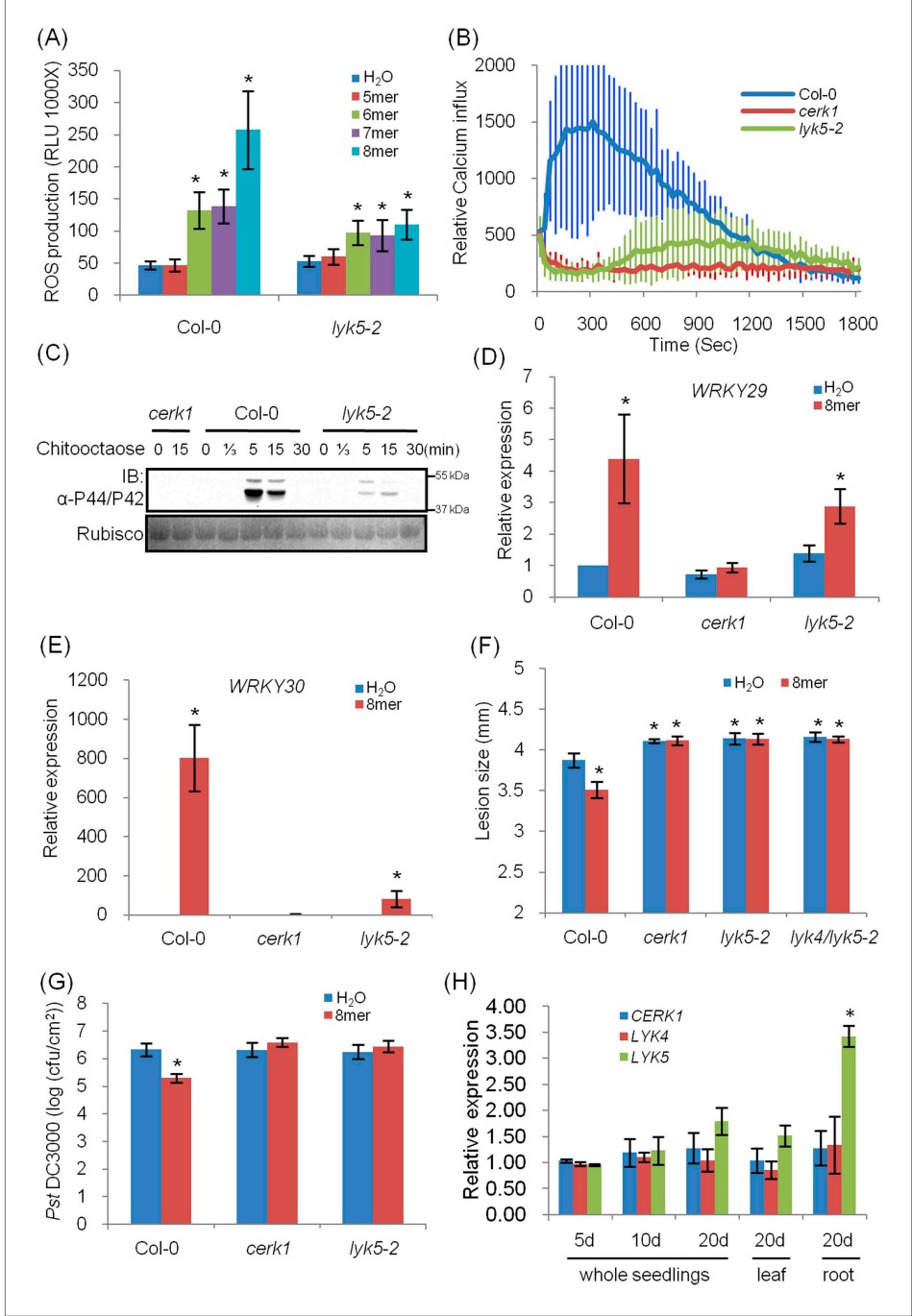

**Figure 1**. *Atlyk5* mutant plants are defective in chitin-triggered immune responses. (**A**) ROS production was measured from Col-0 wild-type and *Atlyk5-2* mutant plants for 30 min after treatment with different chitin oligomers. 5mer: chitopentaose, 6mer: chitohexaose, 7mer: chitoheptaose, and 8mer: chitooctaose. Data are mean ± SE (n = 8). Asterisks indicate significant difference relative to H$_2$O treated Col-0 wild-type plants. (p < 0.01, Student's *t* test). (**B**) Calcium influx in the wild-type, *Atcerk1* and *Atlyk5-2* mutant plants expressing aequorin was recorded for 30 min after chitooctaose treatment. (**C**) MAP kinase phosphorylation after chitooctaose treatment was detected by immunoblot using anti-P44/P42 antibody. (**D**) *AtWRKY29* (At4g23550) and (**E**) *AtWRKY30* (At5g24110) gene expression was analyzed using qRT-PCR in the wild-type, *Atcerk1* and *Atlyk5-2* mutant plants with or without treatment with chitooctaose, 8mer. *UBQ10* (At4g05320) was used a control. Data are mean ± SE (n = 3). Asterisks indicate significant difference relative to H$_2$O treated Col-0 wild-type plants. (p < 0.01, Student's *t* test). (**F**) 4-week-old leaves from Col-0 wild-type, *Atcerk1*, *Atlyk5-2*, and *Atlyk4/lyk5-2* mutant plants were inoculated with
*Figure 1. Continued on next page*

*Figure 1. Continued*

*Alternaria brassicicola* 24 hr after hand-infiltration with $H_2O$ or 1 µM chitooctaose. The diameter of the lesion area was measured 4 days after inoculation. Data are mean ± SE (n = 12). Asterisks indicate significant difference relative to $H_2O$ treated Col-0 wild-type plants. (p < 0.05, Student's *t* test). (**G**) Leaf populations of *Psuedomonas syringae* pv. tomato DC3000 3 days after inoculation. 4-week-old plants were either pretreated with $H_2O$ or 1 µM chitooctaose 24 hr before inoculation with *P. syringae*. Data are mean ± SE (n = 9). Asterisk indicates T-test significant difference compared with $H_2O$-treated Col-0 plants at p < 0.05, Student's *t* test. (**H**) *AtCERK1, AtLYK4* and *AtLYK5* gene expression in different plant ages and plant tissue. RNA from whole seedling of 5 day, 10 day, 20 day old plants and leaf and root tissues from 20 day old plants were used for reverse transcript and qRT-PCR was performed using specific primers. Data are mean ± SE (n = 3). Asterisks indicate significant difference relative to chitiooctaose treated Col-0 wild-type plants (p < 0.01, Student's *t* test).

The following figure supplements are available for figure 1:

**Figure supplement 1**. Arabidopsis LYK gene family.

**Figure supplement 2**. Chitin response in Ler *lyk5-1* mutant plants.

**Figure supplement 3**. Characterization of *Atlyk5* mutant plants.

**Figure supplement 4**. *WRKY33* and *WRKY53* gene expression in *Atlyk5-2* mutant plants.

**Figure supplement 5**. Chitin-induced ROS production in five *lyk* mutant plants.

MPK6 are specifically phosphorylated upon chitin elicitation in Col-0 wild-type but not in *Atcerk1* mutant plants. Significant reduction in MPK phosphorylation was detected in *Atlyk5-2* mutant plants after chitin treatment (*Figure 1C*). Consistent with these findings, the *Atlyk5-2* mutant plants showed an intermediate response with regard to chitin-induced *AtWRKY29, AtWRKY30, AtWRKY33,* and *AtWRKY53* expression compared to the Col-0 wild-type and the *Atcerk1* mutant plants (*Figure 1D,E*, *Figure 1—figure supplement 4*). Both *Atcerk1* and *Atlyk5-2* mutant plants showed increased susceptibility to the fungal pathogen *Alternaria brassicicola* compared with Col-0 wild-type plants. Pretreatment with chitooctaose enhanced resistance to *A. brassicicola* in wild-type plants but not in *Atcerk1* or *Atlyk5-2* mutant plants (*Figure 1F*). Untreated *Atcerk1* and *Atlyk5-2* mutant plants showed wild-type levels of resistance to the bacterial pathogen *Pseudomonas syringae* pv. tomato DC3000. However, only the wild-type showed increased bacterial resistance when plants were pretreated with chitooctaose to induce MTI (*Figure 1G*). In order to confirm that the loss of chitin response was due to the *AtLYK5* mutation, transgenic plants expressing the full-length *AtLYK5* gene were generated under control of its native promoter in the *Atlyk5-2* mutant genetic background (*Figure 2—figure supplement 1*). Expression of AtLYK5 in *Atlyk5-2* mutant plants complemented all of the chitin-induced responses, including ROS production and MAPK phosphorylation (*Figure 2—figure supplement 1*). These data indicate that AtLYK5 is essential for a strong response to chitin elicitation. The response of all five *Atlyk* mutant plants was tested based on chitin-induced ROS prodction (*Figure 1—figure supplement 5*), confirming that AtCERK1, AtLYK4 and AtLYK5, but not AtLYK2 or AtLYK3 are involved in chitin signaling.

## AtLYK5 shares overlapping function with AtLYK4 in mediating the chitin response

Analysis of the *Atlyk5* mutant plants showed some residual response to chitin. Previously, we reported that AtLYK4 is also required for chitin elicitation (*Wan et al., 2012*). Phylogenetic analysis shows that AtLYK5 and AtLYK4 are in the same branch (*Figure 1—figure supplement 1*). Therefore, we examined the possibility that AtLYK4 may provide some functional redundancy for the loss of AtLYK5; thus, explaining the low level response of the *Atlyk5* mutants to chitin. To address this hypothesis, mutant plants defective in both AtLYK4 and AtLYK5 were generated through crossing (*Figure 2—figure supplement 1*) and tested for their response to chitin. Similar to the *Atcerk1* mutants, plants mutated in AtLYK4 and AtLYK5 lost all tested responses to chitin, including ROS production and MAPK phosphorylation, as well as resistance to the fungal pathogen *A. brassicicola* (*Figure 1F* and *Figure 2*). These

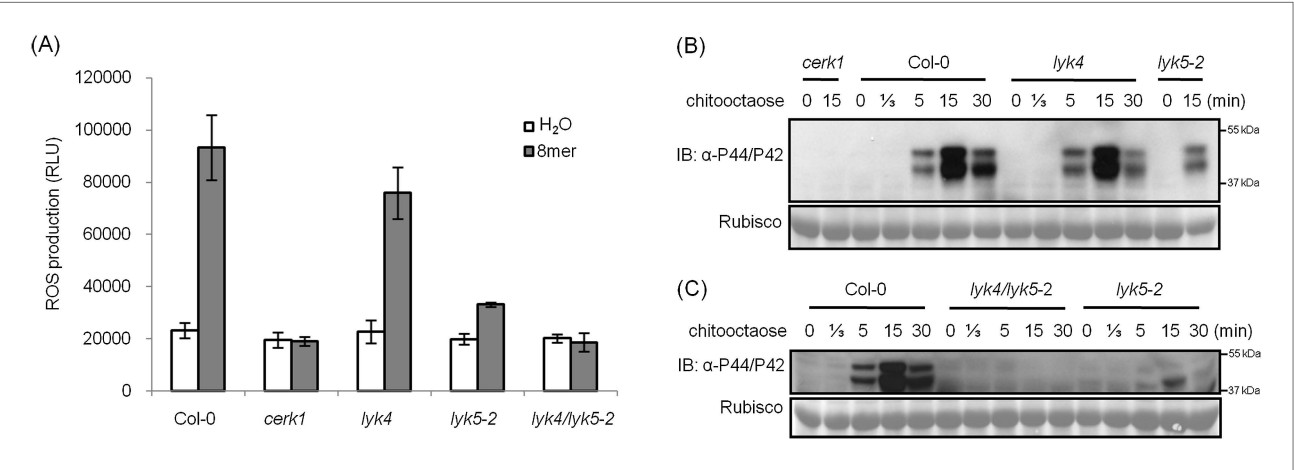

**Figure 2**. AtLYK5 has overlapping function with AtLYK4. (**A**) ROS production was measured from Col-0 wild-type, *Atyk4*, *Atlyk5-2*, and *Atlyk4/lyk5*-2 mutant plants for 30 min after treatment with $H_2O$ (as control) or 1 μM chiooctaose. 8mer: chitooctaose. Data are mean ± SE (n = 8). (**B**) and (**C**) Western blot of total protein extracts from plants treated with 1 μM chitooctaose. Protein was separated by SDS-PAGE gel and visualized using anti-P44/P42 antibody. Upper panel in each figure shows phosphorylated MPK3 and MPK6, lower panel shows similar loading of each lane stained with ponceau S solution.

The following figure supplements are available for figure 2:

**Figure supplement 1**. Complementation of *Atlyk5-2* mutant plants.

**Figure supplement 2**. Tissue-specific expression of *AtCERK1*, *AtLYK4*, and *AtLYK5*.

genetic studies suggest that AtLYK5 and AtCERK1 are essential for a strong plant chitin response, while AtLYK4 can partially compensate for the loss of AtLYK5.

Because AtCERK1, AtLYK5, and AtLYK4 are all involved in the chitin response, we measured their transcriptional level in different plant tissues at different plant ages using quantitative PCR (qRT-PCR). At different growth stages of 5, 10, and 20 days after germination, as well as leaf and root tissues from 20-day-old plants, the transcript levels of three genes were similar. However, in the root tissue tested, *AtLYK5* expression was higher than *AtCERK1* and *AtLYK4*, while both showed similar expression levels in roots (*Figure 1H*). These data are consistent with the results predicted by the AtGenExpress Visualization Tool (AVT) showing that *AtLYK5* is co-expressed with *AtCERK1*, with some variation in the root (*Figure 2—figure supplement 2* or online at http://jsp.weigelworld.org/expviz/expviz.jsp).

Previous reports showed that both AtCERK1 and AtLYK4 are localized on the plasma membrane (*Miya et al., 2007*; *Wan et al., 2012*). The AtLYK5 was fused with c-terminal GFP and transiently expressed under the CaMV *35S* promoter in *Nicotiana benthamiana*. Confocal microscopy showed that AtLYK5-GFP co-localized with the plasma membrane dye, FM4-64, and western blots showed the correct size of the AtLYK5-GFP protein (*Figure 2—figure supplement 2*). These results indicate that together with AtCERK1 and AtLYK4, AtLYK5 is a membrane-localized protein.

## AtLYK5 binds to chitin with higher affinity than AtCERK1

Given the low reported affinity of AtCERK1 for chitooctaose (*Liu et al., 2012b*), we tested the ability of AtLYK5 to bind to chitin. HA-tagged versions of each of the five AtLYK proteins were expressed in *Arabidopsis* protoplasts and chitin-magnetic beads were used to pull down any chitin binding proteins. As shown in *Figure 3—figure supplement 1*, besides AtCERK1, only AtLYK4 and AtLYK5 were also pulled down by chitin beads. The binding with AtLYK5 was strongly inhibited by chitoheptaose and chitooctaose, whereas the binding with AtCERK1 or AtLYK4 was only slightly reduced by the same competitors, consistent with AtLYK5 having a higher binding affinity for chitooctaose relative to AtCERK1.

In order to further investigate this possibility, the chitin binding ability of the extracellular domains of AtCERK1 and AtLYK5 were measured using isothermal titration calorimetry (ITC). As shown in *Figure 3*, the binding affinity of AtLYK5 for chitooctaose was measured ($K_d$ = 1.72 μM), which is roughly

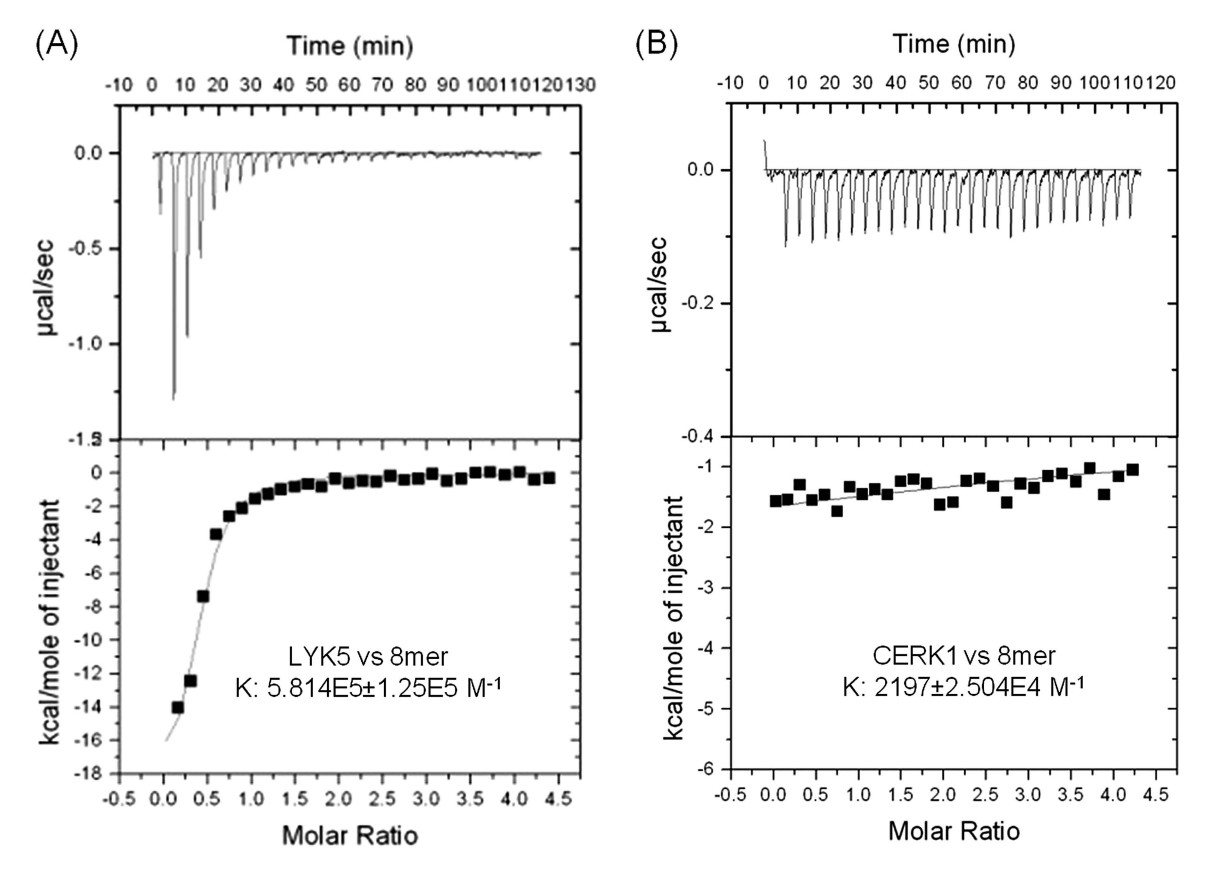

**Figure 3**. AtLYK5 shows stronger chitin binding affinity than AtCERK1. The binding affinity of AtLYK5 (**A**) and AtCERK1 (**B**) to chitooctaose $(GlcNAc)_8$ was measured using isothermal titration calorimetry (ITC). Proteins were purified from *E. coli*. Upper panels and lower panels indicate raw data and integrated heat values, respectively.

The following figure supplement is available for figure 3:

**Figure supplement 1**. AtLYK5 has chitooctaose binding affinity.

200-fold higher than measured for AtCERK1 under the same conditions ($K_d$ = 455 µM) (***Figure 3A,B***). In control experiments, no chitin binding affinity was detected using buffer titrated with chitooctaose (***Figure 3—figure supplement 1***). As expected, AtLYK5 showed no binding affinity for elicitor-inactive chitotetraose (***Figure 3—figure supplement 1***). As a positive control, wheat germ agglutinin (WGA) showed strong chitin binding affinity ($K_d$ = 1.6 µM; ***Figure 3—Figure supplement 1***). Therefore, the data indicate that AtCERK1 shows very low affinity for chitooctaose, while AtLYK5 shows an affinity very close to the well-characterized chitin binding protein WGA.

## Tyr-128 and Ser-206 residues are important for the AtLYK5-mediated chitin response

The AtCERK1 crystal structure predicted chitooctaose binding to the second LysM motif of the extracellular domain leading to homodimerization and kinase activation (***Liu et al., 2012b***). However, AtCERK1 appears to be a very weak chitin binding protein raising questions as to the biological relevance of the AtCERK1 homodimer model. Therefore, in order to predict the chitin binding site(s) within the AtLYK5 extracellular domain, a computational model of the AtLYK5 ectodomain was built by homology modeling against the known crystal structure of the fungal ECP6 (***Sanchez-Vallet et al., 2013***), a LysM effector protein, which binds chitin with very high affinity (binding at pM levels; ***Figure 4A,B*** and ***Figure 4—figure supplement 1***). Based on the docking model of AtLYK5 with chitooctaose, the binding affinity was calculated at −8.9 kcal mol$^{-1}$ (***Figure 4A,B***), a value comparable to

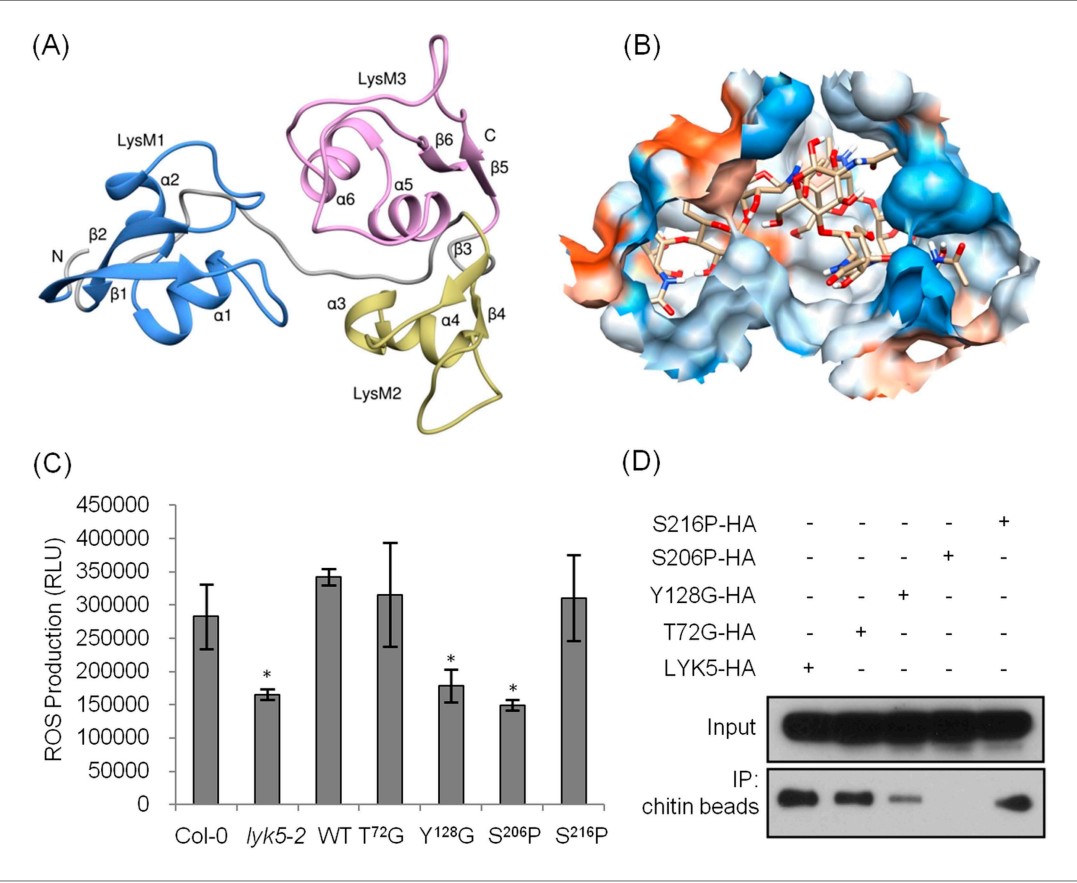

**Figure 4**. Tyr-128 and Ser-206 are important for AtLYK5-mediated chitin response. (**A**) A computational ribbon structure of the AtLYK5 ectodomain was built based on crystal structure of fungal ECP6. The model shows the three AtLYK5 LysM domains, i.e. LysM1-3. Each LysM domain contains two beta strands and two helixes interconnected via loops. (**B**) The binding affinity was calculated at −8.9 kcal mol$^{-1}$. The binding site was formed by 3 LysM motifs. Green lines depict hydrogen bonds formed between ligand atoms and their corresponding residues atoms. (**C**) Reactive oxygen species (ROS) was measured within 30 min after chitin treatment. The *AtLYK5* wild-type gene or versions with specific point mutations were transformed into *Atlyk5-2* mutant plants. Eight individual transgenic plants were used for this measurement. Data are mean ± SE. Asterisks indicate significant difference relative to H$_2$O treated Col-0 wild-type plants. (p < 0.01, Student's *t* test). (**D**) Chitin binding affinity of AtLYK5 and AtLYK5 mutant proteins as labeled in (**C**) detected by anti-HA antibody. Upper panel shows input of each transgenic plant, lower panel shows western blot after pull down with chitin-magnetic beads.

The following figure supplement is available for figure 4:

**Figure supplement 1**. Computational model of the extracellular domain of AtLYK5.

the computational binding affinity of ECP6 (−9.0 kcal mol$^{-1}$) (***Figure 4—figure supplement 1***). Four residues, that is, Thr-72, Tyr-128, Ser-206, and Ser-216, were predicted to form hydrogen bonds and hydrophobic interactions with chitooctaose based on docking model (***Figure 4—figure supplement 1***). Point mutations were introduced at each of these residues and transgenically expressed in *Atlyk5* mutant plants from the native promoter. As shown in ***Figure 4C***, *AtLYK5$^{S206P}$* and *AtLYK5$^{Y128G}$* transgenic plants could not rescue the *Atlyk5-2* mutant phenotype as measured by chitin-triggered ROS production. In contrast, expression of AtLYK5$^{T72G}$ and AtLYK5$^{S216P}$ mutant proteins in the *Atlyk5-2* mutant plants did restore the chitin response. Consistent with these results, AtLYK5$^{S206P}$ mutant proteins did not bind to chitin beads, while AtLYK5$^{Y128G}$ mutant proteins showed a strong reduction in chitin binding using this same assay (***Figure 4D***). Binding of the AtLYK5$^{T72G}$ and AtLYK5$^{S216P}$ mutant proteins to the chitin beads was similar to wild-type AtLYK5 (***Figure 4D***). These data indicate that residues Tyr-128 and Ser-206 of AtLYK5 are important for chitin binding and that chitin binding is essential for biological activity.

## The association between AtLYK5 and AtCERK1 is induced by chitin elicitation

Given that both AtCERK1 and AtLYK5 are required for a strong chitin response, we hypothesized that AtLYK5 might interact with AtCERK1. Indeed, co-immunoprecipitation (Co-IP) assays showed that AtCERK1-HA strongly interacts with AtLYK5-Myc in the presence of chitin (*Figure 5A*). Chitin-induced association between AtLYK5 and AtCERK1 was much stronger than homodimerization of AtCERK1 (*Figure 5A*). We also tested the chemical specificity of this response using chitin oligomers of increasing length (*Figure 5B*). Chitopentaose which can not trigger immune responses in plants did not induce AtLYK5-CERK1 association, while chitin oligomers (dp ≥ 6), which are strong elicitors, induced the interaction between AtLYK5 and AtCERK1 (*Figure 5B*). We also tested the association between AtLYK4 and AtCERK1 before and after chitin treatment. As shown in *Figure 5—figure supplement 1*, AtCERK1-HA could be co-immunoprecipitated with AtLYK4-Myc; however, this interaction was independent of the presence of chitin.

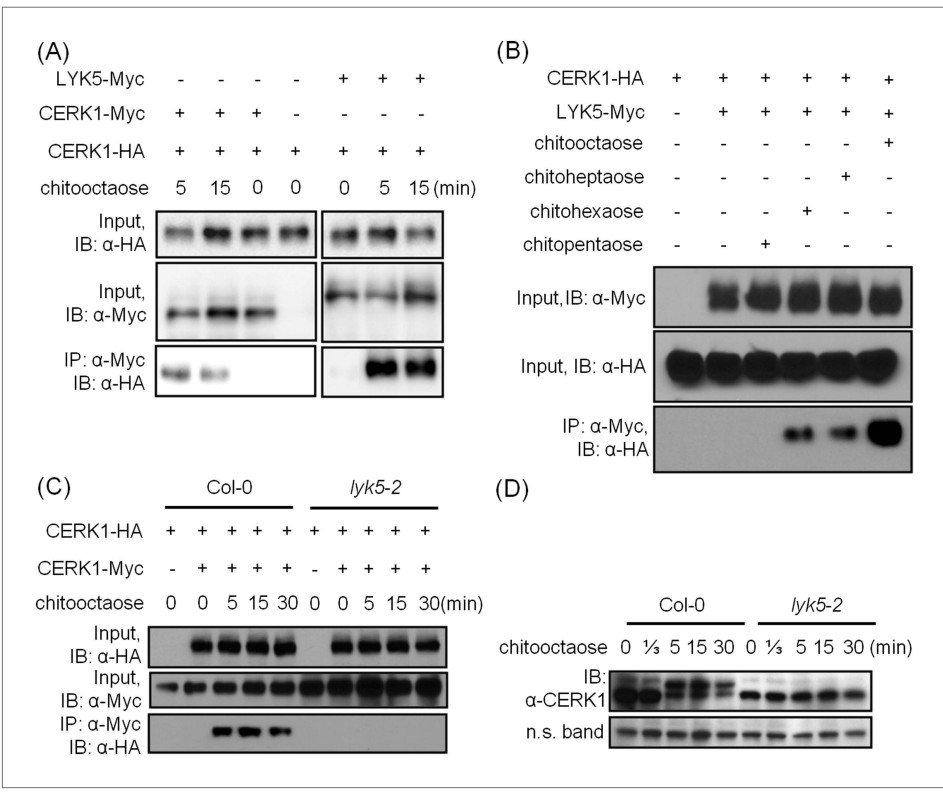

**Figure 5**. AtLYK5 regulates chitin-induced phosphorylation and homodimerization of AtCERK1. (**A**) AtLYK5 associates with AtCERK1 after chitin treatments. HA-tagged AtCERK1 and Myc-tagged AtLYK5 or AtCERK1 were co-expressed in protoplasts made from Col-0 wild-type plants. Protoplasts were harvested with or without the treatment with 1 μM chitooctaose as labeled above. Co-immunoprecipitation was made using anti-Myc antibody. Left panel and right panel are cropped from the same gel. (**B**) The association between AtCERK1 and AtLYK5 is induced by different chitin oligomers. Protoplasts were treated with different chitin oligomers (1 μM) as shown above for 15 min. (**C**) AtLYK5 regulates chitin-induced AtCERK1-AtCERK1 association. HA-tagged AtCERK1 and Myc-tagged AtCERK1 were copexpressed in protoplasts made from Col-0 wild-type or *Atlyk5-2* mutant plants. Protoplasts were harvested with or without the treatment with 1 μM chitooctaose. Co-immunoprecipitation was made using anti-Myc antibody. (**D**) AtLYK5 controls chitin-induced phosphorylation of AtCERK1. Plant leaves from wild-type and the *Atlyk5-2* mutant plants were treated with 1 μM chitooctaose for the time shown above. Anti-AtCERK1 antibody was used to detect the phosphorylation status of AtCERK1 shown as a shift in protein migration. Lower panel shows a non-specific band used to assess similar loading of each lane.

The following figure supplement is available for figure 5:

**Figure supplement 1**. AtLYK4 associates with AtCERK1 before and after chitin treatment.

## AtLYK5 is necessary for chitin-induced AtCERK1 phosphorylation and homodimerization

Previously, it was postulated that chitin-induced AtCERK1 homodimerization leads to AtCERK1 phosphorylation (*Liu et al., 2012b*; *Petutschnig et al., 2010*), required for downstream activation of plant innate immunity. Consistent with this model, co-immunoprecipitation demonstrated AtCERK1 dimerization upon chitin addition in wild-type plants (*Figure 5C*). However, this association disappeared in the *Atlyk5-2* mutant plants (*Figure 5C*). Chitin-induced phosphorylation of AtCERK1 was detected as a protein mobility-shift on SDS-PAGE gels. In contrast, no phosphorylation of AtCERK1 was detected after chitin elicitation of *Atlyk5-2* mutant plants (*Figure 5D* and *Figure 5—figure supplement 1*). These data clearly demonstrate that AtLYK5 is essential for both AtCERK1 dimerization and subsequent activation of protein phosphorylation.

## The AtLYK5 kinase domain is inactive but is required for chitin-induced association with AtCERK1

A comparison was made using the amino acid sequences of the intracellular kinase domain of AtLYK5 and other known receptor kinases, including AtCERK1 (*Miya et al., 2007*; *Wan, et al., 2008*), AtBRI1 (*Li and Chory, 1997*), AtBAK1 (*Li et al., 2002*; *Nam and Li, 2002*), AtLYK4 (*Wan et al., 2012*), and Does not Respond to Nucleotides 1 (AtDORN1) (*Choi et al., 2014*). The analysis showed that several residues generally considered essential for kinase activity are missing from the AtLYK5 sequence (*Figure 6—figure supplement 1*), including those in the P-loop sub-domain I, RD domain in sub-domain VIa, and DFG domain in sub-domain VII (*Figure 6—figure supplement 1*), suggesting that AtLYK5 lacks kinase activity (*Hanks et al., 1988*). In order to test this directly, the AtLYK5 kinase domain was expressed and purified from *Escherichia coli* and used in an in vitro kinase assay. As shown in *Figure 6A*, as a positive control, the AtCERK1 kinase domain showed strong kinase activity as previously reported (*Miya et al., 2007*), while no kinase activity was detected using the AtLYK5 kinase domain. In order to investigate the role of AtLYK5 kinase activity in vivo, wild-type AtLYK5, a mutant AtLYK5$^{K395E}$ (mutation of Lys to Glu predicted to disrupt ATP binding ability), and mutant AtLYK5$^{\Delta KD}$ (deletion of kinase domain) proteins were transgenically expressed in *Atlyk5-2* mutant plants from the native promoter (*Figure 6—figure supplement 1*). The data show that expression of AtLYK5 or AtLYK5$^{K395E}$ could complement the *Atlyk5-2* mutant phenotype, as measured by chitin-triggered ROS production, MAPK phosphorylation, and AtCERK1 phosphorylation, whereas transgenic expression of AtLYK5$^{\Delta KD}$ did not complement the *Atlyk5-2* mutant (*Figure 6B–E*). These results indicate that although AtLYK5 kinase activity is not required for chitin signaling, the intracellular kinase does have a function, which may include mediating protein–protein interactions. Indeed, the AtLYK5$^{\Delta KD}$ mutant protein could not be co-immunoprecipitated with AtCERK1 after chitin treatment, while AtCERK1 interacted with AtLYK5$^{K395E}$ and wild-type AtLYK5 normally (*Figure 6F*). These data strongly demonstrate that the kinase domain of AtLYK5 is necessary for the association of AtLYK5 and AtCERK1.

## AtLYK5 homodimerizes before and after chitin elicitation

Receptor homodimerization or oligomerization is a common mechanism for ligand-mediate receptor activation (*Jiang and Hunter, 1999*; *Mellado et al., 2001*; *Pang and Zhou, 2013*; *Stock, 1996*). Therefore, we tested whether AtLYK5 could form homodimers and whether this required AtCERK1 and/or chitin treatment. As shown in *Figure 7*, AtLYK5 homodimers were detected even in the absence of chitin and this association was independent of the presence of AtCERK1 (*Figure 7A*). AtLYK5 homodimers were also detected in vivo (*Figure 7B*). In the presence of dithiothreitol, a reducing reagent that disrupts disulfide bonds, AtLYK5 proteins became monomers (*Figure 7B*). In addition, we did not observe any molecules larger than AtLYK5 dimers, suggesting no oligomerization of AtLYK5 (*Figure 7B*).

## Discussion

The current model for AtCERK1 function indicates that this protein directly binds long chain chitooligosaccharides (dp ≥ 6), leading to homodimerization, kinase activation and downstream induction of MTI (*Bohm et al., 2014*; *Kadota et al., 2014*; *Liu et al., 2012b*). The problem with this model is that it does not account for the relatively low binding affinity of AtCERK1 for chitin and the fact that mutations (AtCERK1$^{A138H}$) predicted to disrupt chitin binding did not block AtCERK1 autophosphorylation (*Liu et al., 2012b*). These discrepancies can now be explained by the interaction of AtCERK1 with AtLYK5, which binds chitin with a significantly higher affinity than AtCERK1. Indeed, AtLYK5 is required

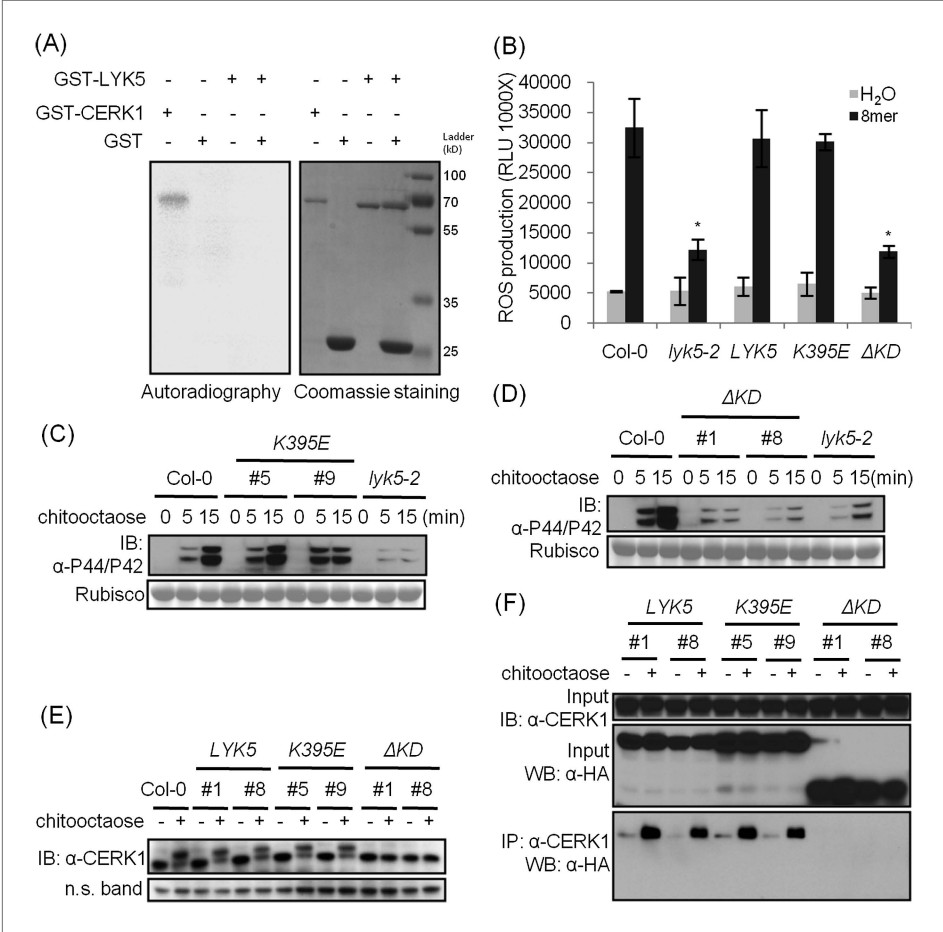

**Figure 6**. The kinase domain of AtLYK5 is critical for chitin signaling. (**A**) In vitro kinase activities of AtCERK1 (255–617 aa) and AtLYK5 (309–664 aa) were measured by incorporation of γ-[$^{32}$P]-ATP. Left panel shows autoradiography, and right panel shows gel stained with coomassie brilliant blue. (**B–F**) Plant tissues were harvested before (−) or 15 min after (+) treatment or at the time point shown in each figure of treatment with 1 μM chitooctaose. (**B–E**) AtLYK5$^{K395E}$ but not AtLYK5$^{ΔKD}$ (1–320 aa) complemented the *Atlyk5-2* mutant as determined by chitin-triggered ROS production. Asterisks indicate significant difference relative to H$_2$O treated Col-0 wild-type plants. (Data are mean ± SE (n = 8), p < 0.01, Student's *t* test), MPK phosphorylation, and chitin-induced AtCERK1 phosphorylation. Upper panel of each figure show immunoblot data, lower panel shows either rubisco band stained with ponceau S solution (**C** and **D**) or a non-specific band (**E**) to show similar loading of each lane. Plant tissues were harvested before (−) or after (+) 15 min treatment with 1 μM chitooctaose. (**F**) AtLYK5$^{K395E}$ but not AtLYK5$^{ΔKD}$ coimmunoprecipitates with AtCERK1 after chitin elicitation. Co-IP was made using anti-AtCERK1 antibody with proteins from transgenic Arabidopsis *Atlyk5-2* mutant plants expressing either AtLYK5, or AtLYK5$^{K395E}$ or AtLYK5$^{ΔKD}$. 8mer: chitooctaose, *AtLYK5*, *K395E*, and *ΔKD* indicate transgenic Arabidopsis expressing AtLYK5, AtLYK5$^{K395E}$, and AtLYK5$^{ΔKD}$, respectively. Different number indicates different transgenic lines used in this study.

The following figure supplement is available for figure 6:

**Figure supplement 1**. AtLYK5 is a kinase inactive protein.

for AtCERK1 dimerization and also kinase activation. The data suggest that AtLYK5 exists in the cell, in the absence of chitin, as a homodimer. Chitin binds to AtLYK5 leading to its association with AtCERK1, dimerization and kinase activation. In this way, chitin signaling is transduced downstream through AtCERK1 kinase activity.

Our model, together with the studies from rice, suggests that AtCERK1 is not the primary chitin receptor. In rice, one long-chain chitooligosaccharide is sandwiched between two OsCEBiP monomers. Chitin treatment induces the association of OsCEBiP with OsCERK1, which activates the OsCERK1

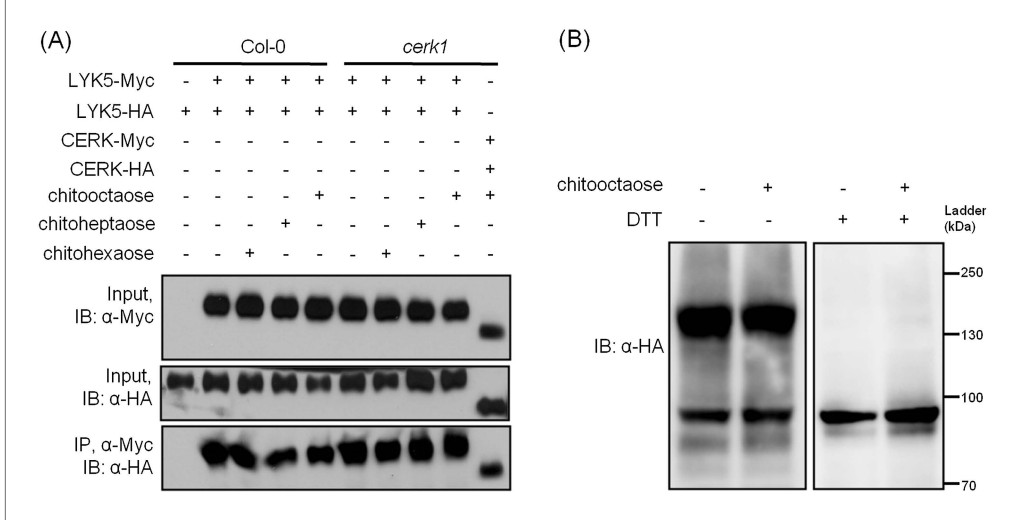

**Figure 7**. AtLYK5 forms a homodimer. (**A**) Homodimeriztion of AtLYK5 is independent on the presence of CERK1 or chitin elicitation. AtLYK5-HA and AtLYK5-Myc, or AtCERK1-HA and AtCERK1-Myc were co-expressed in protoplasts made from Col-0 and *Atcerk1* mutant plants. Protoplasts were harvested before (−) or 15 min after (+) treatment with 1 μM different chitin oligomers. Co-immunoprecipitation was made using anti-Myc antibody. (**B**) Dithiothreitol (DTT) treatment converts AtLYK5 dimer to monomer. Crude protein was extracted from transgenic *Arabidopsis* expressing AtLYK5-HA in *Atlyk5-2* mutant plants. Plant tissues were harvested before (−) and 15 min after (+) treatment with 1 μM chitooctaose. Crude proteins from these tissues were boiled for 5 min before (−) or after (+) adding 50 mM DTT. Left panel and right panel of the immunoblot detected with anti-HA antibody are from the same gel.
The following figure supplement is available for figure 7:

**Figure supplement 1**. A possible working model of chitin receptor in Arabidopsis.

intracellular kinase domain (*Hayafune et al., 2014*; *Shimizu et al., 2010*). However, in *Arabidopsis*, the three CEBiP-like proteins are not required for chitin-triggered innate immunity (*Shinya et al., 2012*; *Wan et al., 2012*). Instead, AtLYK5 appears to play a similar role as CEBiP as a chitin-binding receptor to mediate AtCERK1 activation through ligand-induced association. These two systems are very similar since both OsCEBiP and AtLYK5 lack the intracellular kinase function. OsCEBiP has no intracellular kinase domain, while AtLYK5 contains an intracellular kinase domain that lacks activity. However, the AtLYK5 intracellular kinase domain does appear to have biological function, likely by mediating protein–protein interactions. It is likely that the chitin receptor complex functions as a heterotetramer in both rice (*Hayafune et al., 2014*) and *Arabidopsis* (*Figure 7—figure supplement 1*), although this needs to be confirmed. Such an oligomer complex could explain why in vitro receptor binding affinities (~1 μM) do not correlate well with the measured responses of plants to chitin elicitation at nM concentrations (*Zhang et al., 2002*). An oligomeric AtCERK1 receptor complex is also consistent with the finding that AtLYM1 and AtLYM3 (*Willmann et al., 2011*) are necessary for PGN-induced MTI. All these studies suggest that the primary function of AtCERK1/OsCERK1 is not chitin binding per se, but to serve as a receptor scaffold for interaction with chitin binding co-receptors and, most importantly, to provide intracellular kinase activity.

Based on ITC data, AtLYK5 has similar chitooctaose binding affinity to that of wheat germ agglutinin (WGA) while, under the same conditions, AtCERK1 showed extremely low chitooctaose binding activity. Previously published data, also using ITC, measured the chitopentaose binding affinity of WGA in the low μM range, while the binding affinity ($K_d$) to shorter chitin oligomers $(GlcNAc)_{2-4}$ was significantly lower (i.e., mM level; *Asensio, et al., 2000*). ITC is one of the best methods to study protein–protein interactions or protein-ligand association. Since ITC detects protein–protein or protein-ligand interaction directly which is label-free, the binding affinity value might be slightly different from other methods. For example, in the case of WGA, ITC values ($K_d$) for binding to chitobiose and chitotriose were 1.6 mM and 117 μM, respectively, (*Asensio, et al., 2000*), while similar measurements using surface plasmon

resonance (SPR) gave Kd values of 165 µM and 45 µM, respectively (*Lienenmann et al., 2009*). Published reports for the chitin binding affinity for AtCERK1 also vary widely. For example, (*Iizasa et al., 2010*) measured the affinity of AtCERK1 for chitin on a solid surface (commercial chitin magnetic beads) and reported a value ($K_d$) of 82 nM. It is difficult to reconcile this number with other values obtained by ITC. One possible explanation is that the use of a solid surface promoted oligomerization of AtCERK1 resulting in an enhancement of binding affinity. In contrast, using ITC, the binding affinity ($K_d$) of AtCERK1 for chitotetrose, chitopentaose, and chitooctaose was measured as 159 µM, 66 µM, and 45 µM, respectively (*Liu et al., 2012b*). However, these studies were performed with significantly higher protein and ligand concentration (i.e., 0.1 mM AtCERK1 protein vs 4 mM chitotetraose or chitopenta-ose, and 2.4 mM chitooctaose) than used in our study. Again, such high protein concentrations could have promoted oligomerization of the AtCERK1 protein, which would affect the binding affinities.

*Arabidopsis* has five LYK genes. We can now assign biological function to four of these proteins. AtLYK3 appears to be involved in recognizing short chain (dp = 4–5) lipo-chitooligosaccarides and chitooligosaccharides (*Liang et al., 2013*). However, this recognition leads to a suppression of MTI. The role of AtLYK2 remains unknown but it should be noted that this gene shows very low expression in all tissues examined (*Liang et al, 2014*). AtCERK1/AtLYK1, AtLYK4 and AtLYK5 all appear to mediate the plant response to chitin elicitation leading to MTI. Previously, we reported that the Ler ecotype *Atlyk5-1* mutant was not defective in chitin-induced MTI. However, our current data show that this mutant likely represents a weak mutant allele, perhaps due to the 3′ position of the transposon insertion. The Ler ecotype is also not ideal for studying chitin signaling since it shows a generally weaker response compared to Col-0. In contrast, the Col-0 ecotype *Atlyk5-2* mutant showed a reduced response to chitin treatment in all assays tested, including pathogen response. The phenotype of the *Atlyk5-2* mutant was reversed by genetic complementation with the wild-type AtLYK5 gene.

Chitin-induced MTI requires long chain chitin oligomers (dp ≥ 6) (*Miya et al., 2007*; *Wan et al., 2008*, *2012*). The functions of AtLYK4 and AtLYK5 are partially redundant in that single mutants retain residual responses to chitin, while *Atlyk4/Atlyk5-2* double mutant plants show a complete lack of response to chitin, similar to the *Atcerk1* mutants. However, AtLYK5 appears to have the predominant role in chitin elicitation, as judged by the relative strength of the *Atlyk4* and *Atlyk5-2* mutant pheno-types. There are biochemical differences in the function of AtLYK4 and AtLYK5; for example, AtLYK4 interacts with AtCERK1 independently of the presence of chitin, while AtLYK5-AtCERK1 interaction is chitin dependent. The biological significance of these differences and the functional redundancy of AtLYK4 and AtLYK5 are currently unclear. One possibility is that AtLYK4 also functions as a chitin binding receptor to mediate the plant chitin repsonse.

Receptor homodimerization or oligomerization and subsequent phosphorylation are common mech-anisms for ligand-mediate receptor activation. *Arabidopsis* examples include AtBRI1 homodimeriza-tion and heterodimerization with AtBAK1 (*Li et al., 2002*; *Nam and Li, 2002*; *Santiago, et al., 2013*; *Sun, et al., 2013a*; *Wang et al., 2005*). AtFLS2 appears to exist in the cell as a homodimer before and after ligand perception and then associates with AtBAK1 upon flagellin treatment (*Albert, et al., 2013*; *Chinchilla et al., 2007*; *Heese et al., 2007*; *Sun et al., 2012*; *Sun, et al., 2013b*). The sug-gested mechanism is that ligand perception triggers autophosphorylation between the homodimer or trans-autophosphorylation between/among hetero-interacting proteins to initiate cellular signaling. It appears that all LysM receptors may function as a protein complex. For example, in leguminous plants, the LysM domain, Nod factor receptors (e.g., *Lotus japonicas* Nod factor receptor 1 and 5; LjNFR1 and LjNFR5) function as a heterodimer to mediate high affinity binding to the lipo-chitooligosaccharide Nod factor (*Broghammer et al., 2012*; *Gust et al., 2012*; *Madsen et al., 2011*; *Radutoiu et al., 2003*). It should be emphasized that this model for Nod factor binding is very similar to that of the *Arabidopsis* chitin receptor. In both cases, one LysM RLK (CERK1/NFR1) has an active intracellular kinase domain but interacts with a second LysM RLK (LYK5/NFR5) that lacks intracellular kinase activity. This similarity underlines the now well recognized evolutionary link between chitin, Nod factor and mycorrhizal (Myc) factor recognition (*Liang et al., 2014*).

## Materials and methods

### Plant material and treatment

*Arabidopsis* mutant plants *Atcerk1* (GABI-KAT 096F09), *Atlyk2* (SAIL_318C08), At*lyk3* (SALK_140374), *Atlyk4* (CS850683), *Atlyk5-2* (SALK_131911C), and wild-type Col-0 plants were used in this study.

Typically, 4-week-old *Arabidopsis* plants grown in a condition of 16 hr light/8 hr dark cycle at 22–23°C were used for diverse treatments. For chitin treatment, usually 1 µM chitooctaose (Sigma, St Louis, MO), unless otherwise mentioned, was used to treat plant tissues. Homozygous *Atlyk5-2* mutant and *Atlyk4/Atlyk5-2* double mutants plants were genotyped using primers listed in *Supplemental file 1*.

## Gene cloning and plasmid construction

All primers used for gene cloning are listed in *Supplemental file 1*. The full-length cDNAs of *AtLYK2*, *AtLYK3*, *AtLYK4*, and *AtLYK5* were amplified using the template made by *Wan et al. (2012)* and cloned into pDONR-Zeo plasmid by BP cloning. The resultant plasmid was then used for LR cloning with destination plasmids pUC-GW14 and pUC-GW17 (*Cao et al., 2013*). The resultant plasmids were used for protein expression in *Arabidopsis* protoplasts.

Two genomic DNA fragments, containing the 1.5 kb promoter of *AtLYK4* and *AtLYK4* coding region up to the stop codon and the 1.8 kb promoter of *AtLYK5* and *AtLYK5* coding region up to the stop codon, and cDNA of *AtLYK5*, were individually amplified with *pfu* Ultra II HF (Agilent Technologes, Santa Clara, CA) and cloned into the pDONR-Zeo vector using BP clonase (Invitrogen, Carlsbad, CA). The resulting plasmids were then recombined into the destination binary vector pGWB13 or pGWB5 (*Nakagawa et al., 2007*) for *Arabidopsis* transformation or transient expression in *N. benthamiana*.

Kinase domains of AtLYK5 (309–664 aa) and AtCERK1 (255–617 aa) were amplified and inserted into pGEX 5X-1 between *Eco*R I and *Xho* I and between *Eco*R I and *Sal* I, separately, and transformed into BL21 (DE3) for recombinant protein expression.

## *Arabidopsis* protoplast preparation and transformation

*Arabidopsis* protoplasts were prepared from 4-week-old plants according to the protocol described by *Yoo et al. (2007)*. For protein expression, protoplasts (200 µl, about $2 \times 10^5$ cells) were transfected with 20 µg plasmids. For co-immunoprecipitation assays, protoplasts (1 ml; ~$10^6$ cells) were transfected with 100 µg plasmid. After incubation in a growth chamber at 23°C overnight (14–16 hr), the transfected protoplasts were treated with chitooctaose for the times shown in the figure legends and frozen in liquid nitrogen and stored in −80°C for further use.

## Co-immunoprecipitation assay

Samples from either protoplasts or plant tissues were lysed in a buffer containing 50 mM Tris (PH 7.6), 150 mM NaCl, 0.5% Triton X-100 and 1 × protease inhibitor (Sigma, MO). The resulting extract was centrifuged at 14,000 rpm for 15 min at 4°C. Either anti-Myc (Covance, Princeton, New Jersey) or anti-AtCERK1 antibody was used for CoIP experiments according the method described by *Cao et al. (2013)*. Anti-AtCERK1 antibody was made based on the peptide N'-CNFQNEDLVSLMSGR-C' located at c-terminal of AtCERK1 by GenScript Company (Piscataway, NJ).

## Chitin-induced CERK1 phosphorylation

1 µM chitooctaose or $H_2O$ was hand-infiltrated into leaves of different plants. At the time points shown in the figures, leaves were harvested and frozen in liquid nitrogen and ground using a Bead Ruptor Homogenizer (Omni, Kennesaw, GA). Samples were placed on ice for 30 min during the lysis. After centrifuging at 13,000 rpm at 4°C for 15 min, the supernatants were boiled for 5 min in 1 × SDS loading buffer. The phosphorylated AtCERK1 was separated on 7% SDS-PAGE gel at low voltage (60–80 V) for 4 hr or until the protein ladder with 70 kDa reached the bottom of the gel. For dephosphorylation assay, antarctic phosphatase (New England Biolabs, Ipswich, MA) was used for treatment for 15 min at 37°C. AtCERK1 was detected with anti-AtCERK1 antibody.

## ROS production, MAPK phosphorylation, and pathogen assay

ROS production and MAPK phosphorylation assays were performed as described by *Liang et al. (2013)*. The disease assay with *A. brassicicola* was conducted as described by *Wan et al. (2008)*. 1 µM chitooctaose or $H_2O$ (control) was infiltrated into leaves from 4-week-old plants 24 hr before bacterial infiltration. The bacterial pathogen assay was carried out according to the method described by *Cao et al. (2013)*.

## Calcium influx assay

Aequorin transgenic seeds were kindly provided by Dr Marc R Knight (University of Oxford). The *Atlyk5-2* mutant was crossed with Col-0 aequorin to make an *Atlyk5-2* aequorin line. Calcium influx assays were done using the same method described by *Liang et al. (2013)*.

## Purification of GST-fusion protein and in vitro kinase assay

Purification of AtCERK1 and AtLYK5 intracellular domains fused with N-terminal GST tag and in vitro kinase assay were performed as described by *Cao et al. (2013)*. Briefly, cultures of *E. coli* strains harboring plasmid were supplemented with 0.1 mM isopropyl b-D-thiogalactopyranoside (IPTG) at $OD_{600}$ 0.8 at 18°C for 12 hr. The cells were lysed in a buffer containing 1 × PBS (MP Biomedicals, France) supplemented with 1 mM EDTA, 0.1% Triton X-100, 1 mg/ml lysozyme and placed on ice for 30 min with slow shaking before sonication. After centrifuge at 4°C for 30 min, the supernatant was applied to a column containing glutathione Sepharose 4B (GE Healthcare, Milwaukee, WI) for protein purification. The column was washed five-times with 1 × PBS buffer. Recombinant proteins were eluted with 10 mM reduced glutathione. Proteins were dialyzed with a buffer containing 50 mM Tris (PH 7.6), 50 mM KCl, and 10% glycerol. For in vitro kinase assay, proteins were incubated in the buffer [50 mM Tris (PH 7.6), 50 mM KCl, 10 mM $MnCl_2$, 10 mM $MgCl_2$, 10 mM ATP, and 10 mCi γ-[$^{32}$P]-ATP at room temperature for 30 min. Autoradiography was performed using a phosphor screen and a phosphorimager.

## Quantitative reverse transcript PCR (qRT-PCR)

Total RNA was extracted from 10-day-old seedlings using an RNase easy kit (Invitrogen, Grand Island, NY) according to the manufacturer's instructions. qRT-PCR was carried out as previously described by *Tanaka et al. (2011)*. The primers used are listed in *Supplemental file 1*.

## *Arabidopsis* transformation assay

All the binary vectors were electroporated into *Agrobacterium tumefaciens GV3101* (pMP90) and transformed by the floral dip method (*Clough and Bent, 1998*) into *Atlyk5-2* mutant plants. Transgenic plants were selected on half strength MS with 25 mg/l hygromycin after seed surface sterilization described by *Clough and Bent (1998)*.

## Expression of AtLYK5-GFP in *N. benthamiana*

*AtLYK5* cDNA in pDONR-Zeo vector was recombined into binary vector pGWB5 using LR reaction. The resultant plasmid was electroporated into agrobacterial strain GV3101 for transiently expression in *N. benthamiana* according the method described by *Wan et al. (2012)*. 2 days after infiltration, the infiltrated leaf was used to monitor fluorescence signal using confocal microscope.

## Purification of AtCERK1 and AtLYK5 ectodomains

The ectodomain of AtLYK5 (27–278 aa) was amplified and inserted into pMCSG73 using the ligation-independent procedure (PMID: 18988021). A thrombin cleavage site (LVPRGS) was inserted right before the His tag using site-directed mutagenesis PCR. Cultures of *E. coli* BL21, expressing the appropriate plasmid, were supplemented with 0.1 mM IPTG at $OD_{600}$ 0.8 at 18°C for 24 hr. Recombinant protein was purified with Talon metal affinity resin and eluted with 300 mM imidazole. Eluted protein was then incubated with thrombin at 4°C overnight and subsequently dialyzed with (50 mM Tris (PH 8.0) 150 mM NaCl). The resultant protein was purified over Strep-Tactin resin. The ectodomain of AtCERK1 (24–231 aa) was amplified and inserted into pMAL-c2G vector at the *Bam*H I and *Sal* I sites. After expression as described above, the recombinant protein was purified over amylose affinity resin according to the manufacturer's protocol (New England Biolabs, Ipswich, MA).

## Isothermal titration calorimetry

All protein samples including the ectodomains of AtLYK5 and AtCERK1, and wheat germ agglutinin (Sigma, St Louis, MO) were dialyzed against a buffer containing 10 mM HEPES, pH 7.5, 100 mM NaCl. Chitooctaose used for ITC was purchased from Sigma. 20 μM AtLYK5 or AtCERK1 purified protein or 10 μM wheat germ agglutinin (Sigma, St Louis, MO) was titrated separately against either 400 μM chitooctaose or 200 μM chitooctaose at 25°C. As a control, 10 μM AtLYK5 was titrated against 200 μM chitotetraose. Buffer lacking protein was used as control and titrated against 200 μM chitooctaose. The heat from each experiment was measured by MicroCal VP-ITC.

## Modeling and Docking

Homology modeling of the LysM domain of AtLYK5 was performed in the Modeller 9.12 (*Marti-Renom et al., 2000*) using the crystallized structure of ECP6 was used as the template (PDB code 4B8V). The query sequence and template structure alignment was first performed using the Modeller Align

module, and then manually inspected to ensure the best alignment for generating a pool of 2000 models. The best model was selected and analyzed based on Modeller's probability density function (Discrete Optimized Protein Energy score) and the Ramachandran plot (*Laskowski et al., 1993*), and subsequently refined using the Loop refine module of Modeller.

In order to perform docking and calculate a binding affinity, the LysM model and the chitooctaose ligand (NCBI identifier 24978517) were prepared using the MGLTools-1.5.6 software (*Morris et al., 2009*) to satisfy docking requirements such as addition of hydrogen atoms, calculation of partial charge using the AMBER force field (*Cornell et al., 1995*), selection of flexible bonds for the ligand and residues, and adjustment of docking position and grid space (parameters not shown). The putative binding site of the LysM domain model of AtLYK5 was inferred from the template structure. The docking experiment was then performed with the AutoDockVina software (*Trott and Olson, 2010*), and the docking model with the lowest binding energy (expressed in kcal per mol) was selected and visualized in the Chimera software (*Pettersen et al., 2004*). A detailed interaction map between the ligand and surrounding residues was generated by the LigPlus software (*Wallace et al., 1995*).

To calculate the binding affinity of ECP6 to chitooctaose as a positive control, only the crystal model of ECP6 was retrieved for the docking task. The docking experiment with the chitooctaose ligand was performed with the AutoDock Vina software as mentioned above. The docking model with the lowest binding energy was selected and visualized.

## Acknowledgements

We thank Dr Michael Henzl at the University of Missouri for the assistance using ITC and critical comments on this manuscript. We thank Mia Brown and Dr Jason Cooley for their assistance in obtaining CD spectra of AtLYK5 and AtCERK1 protein. We also thank ABRC for providing T-DNA insertion lines.

## Additional information

### Funding

| Funder | Grant reference number | Author |
| --- | --- | --- |
| Rural Development Administration | Next-Generation BioGreen 21 Program Systems and Synthetic Agrobiotech CenterPJ009068 | Gary Stacey |
| Basic Energy Sciences | DE-FG02-08ER15309 | Gary Stacey |
| Biological and Environmental Research | DE-AC02-06CH11357 | Andrzej Joachimiak |
| National Institutes of Health | GM094585 | Andrzej Joachimiak |

The funders had no role in study design, data collection and interpretation, or the decision to submit the work for publication.

### Author contributions

YC, Conception and design, Acquisition of data, Analysis and interpretation of data, Drafting or revising the article; YL, Acquisition of data, Analysis and interpretation of data, Drafting or revising the article; KT, CTN, Acquisition of data, Analysis and interpretation of data; RPJ, AJ, Acquisition of data, Drafting or revising the article; GS, Conception and design, Analysis and interpretation of data, Drafting or revising the article

### Author ORCIDs

Kiwamu Tanaka, http://orcid.org/0000-0001-5045-560X

## Additional files

**Supplementary file**
• Supplementary file 1. Primer sequences used in this study.

## Major dataset

The following previously published dataset was used:

| Author(s) | Year | Dataset title | Dataset ID and/or URL | Database, license, and accessibility information |
|---|---|---|---|---|
| Sanchez-Vallet A, Saleem-Batcha R, Kombrink A, Hansen G, Valkenburg DJ, Thomma BPHJ, Mesters JR | 2013 | Cladosporium fulvum LysM effector Ecp6 in complex with a beta-1,4- linked N-acetyl-D-glucosamine tetramer | http://www.pdb.org/pdb/explore/explore.do?structureId=4B8V | Publicly available at RCSB Protein Data Bank. |

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
