## [Decision Letter]

Thank you for sending your work entitled “Arabidopsis CERK1 is not the primary chitin receptor but acts as a co-receptor with LYK5 to induce plant innate immunity” for consideration at *eLife.* Your article has been favorably evaluated by Detlef Weigel (Senior editor) and 3 reviewers, one of whom, Thorsten Nürnberger, is a member of our Board of Reviewing Editors.

The Reviewing editor and the other reviewers discussed their comments before we reached this decision, and the Reviewing editor has assembled the following comments to help you prepare a revised submission.

All reviewers agree that your manuscript potentially makes a very valuable contribution to *eLife.* In particular, the roles of LYK5 and CERK1 as a genuine chitin receptor/co-receptor pair are exciting findings that could help clarify several issues about major differences between chitin perception systems in different plants. Prior to acceptance you need, however, to address the issues listed below: 1) A major criticism concerns the binding data obtained by ITC. While an affinity constant for WGA (for a positive control) is given, there is no comparable number for chitin affinities of AtCERK1 and AtLYK5. The only statement found is that AtLYK5 has a comparable Kd as WGA, and is 200-fold more sensitive to chitin than AtCERK1. It was unclear to us, however, how the numbers given in Figure 3 or any information in the text support these statements. Here, it is absolutely mandatory to give Kd values and an easily understandable description of how these values were determined. In addition, it should be shown that chitin binding as determined by ITC is ligand-specific. Here, competition assays with elicitor-active and inactive chitin fragments should be provided. In addition, a comprehensive discussion of apparent discrepancies between published chitin binding constants for CERK1 (Liu, 2012 45 uM, [19], 82 nM) and those determined in this study should be provided. This is important as the authors claim that the binding affinity of LYK5 is substantially higher than that of CERK1 and thus makes this protein the genuine chitin receptor.

2) Evidence should be provided that CEERK1 and LYK5 both operate in the same cell types/tissues. This is important as it might rule out that both proteins are chitin receptors with different tissue or cell-type specificities.

3) Figure 1: As previous papers have incorrectly overlooked LYK5 as required for chitin perception, we think this figure needs all of the lyk mutants and a water control. Also; statistics are not clear here or throughout the manuscript. It is not stated what comparison “*” is referred to. Please show the control as 100% and calculate the other samples relative to that. And please explain why you are using at this point wrky29 and wrky30 and not as previously wrky33 and wrky53?

4) Figure 3: The ITC for LYK4 should be done as well, to see if a weaker binding of LYK4 compared to LYK5/CERK1 correlates with the physiological data (ROS, WRKY activation). Overall LYK4 is poorly dealt with in the current manuscript.

---

## [Author Response]

*1) A major criticism concerns the binding data obtained by ITC. While an affinity constant for WGA (for a positive control) is given, there is no comparable number for chitin affinities of AtCERK1 and AtLYK5. The only statement found is that AtLYK5 has a comparable Kd as WGA, and is 200-fold more sensitive to chitin than AtCERK1. It was unclear to us, however, how the numbers given in*
Figure 3
*or any information in the text support these statements. Here, it is absolutely mandatory to give Kd values and an easily understandable description of how these values were determined. In addition, it should be shown that chitin binding as determined by ITC is ligand-specific. Here, competition assays with elicitor-active and inactive chitin fragments should be provided. In addition, a comprehensive discussion of apparent discrepancies between published chitin binding constants for CERK1 (*Liu, 2012 *45 uM,*
[19]*, 82 nM) and those determined in this study should be provided. This is important as the authors claim that the binding affinity of LYK5 is substantially higher than that of CERK1 and thus makes this protein the genuine chitin receptor*.

This is a great point. The Kd value for each protein is now provided in the revised manuscript.

Unfortunately, we could not conduct the competition experiment as you suggest. This is because the ITC method detects protein-ligand interaction based on the differential heat after binding of ligand. Hence, ITC cannot distinguish heat changes produced by LYK5-chitooctaose, or by LYK5- inactive chitin, such as chitotetraose. However, to address criticism, we did conduct direct binding studies of LYK5 with chitotetraose and found no binding (See Figure 3—figure supplement 1). Therefore, as would be expected of a receptor involved in chitin elicitation, LYK5 appears to prefer longer chitin oligomers.

We added a detailed discussion regarding the binding affinity of CERK1. Basically, we feel that the previous studies had technical flaws and, in both cases, binding was done under conditions that would be predicted to favor protein oligomerization which would give an artificially higher binding affinity. In the case of Liu et al this is due to the very high concentrations of protein (0.1 mM) and ligand (4mM or 2.4mM) used in their binding assays. In the case of Iizasa et al, oligomerization would have been favored due to the fact that binding was measured against a solid substrate (chitin beads). Of course, this is conjecture on our part but we believe this is a likely explanation for the apparent discrepancy in the results. Perhaps more importantly, in our case, since we used proteins purified at the same time and assays done under identical conditions, a direct comparison of AtCERK1 and AtLYK5 binding was possible.

*2) Evidence should be provided that CEERK1 and LYK5 both operate in the same cell types/tissues. This is important as it might rule out that both proteins are chitin receptors with different tissue or cell-type specificities*.

Yes, we did real time quantitative RT-PCR using templates from different growth ages and different tissues, and found that CERK1, LYK4 and LYK5 shared a similar expression pattern in all samples tested. The only slight variation was that LYK5 had higher expression in root than CERK1 or LYK4 (See Figure 1 and Figure S1N). Our data is consistent with the expression pattern calculated by an online software AtGenExpress Visualization Tool (AVT): http://jsp.weigelworld.org/expviz/expviz.jsp.

Also, work of another postdoc in the lab has shown that both LYK5 and CERK1 are localized to the plasma membrane, also consistent with their direct interaction.

*3)*
Figure 1*: As previous papers have incorrectly overlooked LYK5 as required for chitin perception, we think this figure needs all of the lyk mutants and a water control. Also; statistics are not clear here or throughout the manuscript. It is not stated what comparison “*” is referred to. Please show the control as 100% and calculate the other samples relative to that. And please explain why you are using at this point wrky29 and wrky30 and not as previously wrky33 and wrky53*?

We retested chitin-triggered ROS production using all five *lyk* mutant plants (see Figure 1—figure supplement 5). These results showed that only *cerk1*, *lyk4* and *lyk5-2* are involved in the chitin response. We also rewrote the section on the statistical analysis and hope that it is now clearer. As we have reported in past talks, we believe that *lyk2* is likely a pseudogene, or at least is not expressed under any conditions tested, while, as we published in Science, LYK3 appears to recognize Nod factor and short chain chitin oligomers. Hence, with this current paper and our past work, we have assigned a function to all of the Arabidopsis LYK proteins that are expressed.

We analyzed the expression of *WRKY33* and *WRKY53* in *lyk5-2* mutant plants. The data showed that the expression of these two genes was greatly reduced after chitin treatment compared with wild-type plants (Figure 1—figure supplement 4).

*4)*
Figure 3*: The ITC for LYK4 should be done as well, to see if a weaker binding of LYK4 compared to LYK5/CERK1 correlates with the physiological data (ROS, WRKY activation). Overall LYK4 is poorly dealt with in the current manuscript*.

Actually, It is a very interesting question. We are also very curious about the binding affinity of AtLYK4.

However, the problem is that expression and purification of the LysM protein is very difficult. Based on our own experience, this is due to an inability to express the proteins in sufficient quantity for such studies. We consider ourselves fortunate that we were able to get enough CERK1 and LYK5 proteins to do the studies described in the paper. Unfortunately, the same was not true of LYK4. Using the same conditions we used for CERK1/LYK4, we tried to express the LYK4 extracellular domain in *E.coli* but obtained poor expression. What protein that was produced was insoluble and efforts to solubilize and refold have, thus far, proven unsuccessful. Therefore, although we would like to address the reviewers and our own curiosity about LYK4, this is going to have to wait until we work out a condition to get sufficient protein for binding studies. Since our current paper focuses on LYK5, these experiments concerning LYK4 are not really pertinent to the story being described. The data support the model that LYK5 is the major chitin binding protein, while LYK4 plays a minor role. We certainly hope to overcome the technical problems with regard to LYK4 expression and, if successful, will likely publish this as a separate story with the main focus on LYK4.